# OpenReview forum: "A Principled Approach to Chain-of-Thought Monitorability in Reasoning Models"
_ICLR.cc/2026/Conference — ICLR 2026 Conference Withdrawn Submission_

### Official Review · Reviewer_5zsH · 2025-10-31

**Soundness:** 3
**Presentation:** 3
**Contribution:** 3
**Rating:** 6
**Confidence:** 3

**Summary:**

This paper introduced a framework for improving the monitorability of CoT reasoning. They identify the current CoT traces are often unfaithful or verbose, and it is hard to control. To address this, this paper propose to use CoT monitorability as a  constraint during optimization. A key contribution is they introduced a prior-guided data generation method: using a prior model to transform unfaithful or lengthy reasoning traces into high-quality and monitorable ones. Experimental results on MMLU-Pro, GSM8K, and MATH500 show the proposed approach improves faithfulness by ~10-22% and reduces the length by ~60%.

**Strengths:**

1. This paper works on a pretty valuable and interesting direction: monitorability in reasoning models. It is important because we need a faithful reasoning model which outputs trustworthy CoT. The area is underexplored in previous literature.
2. The proposed prior-guided data generation method is new and effective, which is a good synergy to RL training recipe.
3. The empirical results are quite strong. It improves the faithfulness by 10% and reduces the length by 60%, although the performance drops a bit (4%).

**Weaknesses:**

1. The proposed prior-guided distillation method heavily relies on an external prior model (e.g., Qwen-7B Instruct) to generate “monitorable” reasoning traces. This suggests that the observed improvement may stem from the prior model’s inductive bias, rather than the generalizability of the proposed framework itself.
2. The faithfulness evaluation depends on judgments made by the llm-as-a-judge approach, lacking objective annotation or inter-rater reliability verification.
3. The authors define monitorability as consisting of faithfulness and conciseness, but do not provide a unified quantitative formulation for this concept.
4. At line 171, the authors used 950 as a threshold to judge the conciseness. I wonder why 950?
5. The experimental design could be improved by introducing more datasets. Testing on MATH500/GSK8K and MMLUPro is not sufficient. Existing models are powerful on more complex tasks like AIME, LiveCodeBench. It is better to test the method on more complex tasks.

**Questions:**

N/A

---

> ### Author Response · Authors · 2025-11-29
>
> > **Weakness 1:**  The proposed prior-guided distillation method heavily relies on an external prior model (e.g., Qwen-7B Instruct) to generate “monitorable” reasoning traces. This suggests that the observed improvement may stem from the prior model’s inductive bias, rather than the generalizability of the proposed framework itself.
>
> **Response to Weakness 1:**
> Below, we clarify why this is **not** the case and provide empirical results demonstrating that our method does *not* rely on a strong teacher for correctness.
>
>
> #### The Teacher Does *Not* Outperform the Student in Reasoning Quality
>
> The prior model that we use—**Qwen 2.5 7B Instruct**—does *not* generate higher-reward or higher-accuracy solutions than the target student model **DeepSeek-R1 Qwen 1.5B**.
>
> To illustrate this, we evaluate the 7B teacher on **MATH500**:
>
> | Model                  | Accuracy | Avg. Tokens |
> |------------------------|----------|-------------|
> | Qwen 2.5 7B Instruct   | 76.4     | 687         |
>
> This demonstrates that the prior is *not* providing superior reasoning or correctness. Therefore, we are **not** relying on a stronger model to “fix” a weaker one.
>
> #### **We Do *Not* Use the Teacher to Generate Correct Answers**
>
> Our method does *not* depend on the teacher’s ability to solve problems or produce faithful reasoning. but the teacher should be good at transformation of $z$ to $z_s$,
>
>
> Thus, the teacher acts as a *rewriter*, not a *reasoner*.
> This distinction is central: **the teacher is not providing new reasoning, only modifying the student’s reasoning.**
>
>
> #### **Evidence: A Much Smaller Teacher Works Equally Well**
>
> To further demonstrate that a strong prior is unnecessary, we repeated the conciseness experiment using an even smaller model: **Qwen 2.5 1.5B Instruct**.
>
> Its standalone performance is significantly weaker:
>
> | Model                    | Accuracy | Avg. Tokens |
> |--------------------------|----------|-------------|
> | Qwen 2.5 1.5B Instruct   | 52.4     | 984         |
>
> Despite this, when used as the teacher for rewriting, the final student achieves **nearly identical results** to the 7B-prior version.
>
>
> #### **Conciseness Experiment Results**
>
> Using Qwen 2.5 7B Instruct as prior (from the main paper)
>
> | Dataset | Accuracy | $f(z)$ |
> |---------|----------|-------------|
> | GSM8K   | 79%      | 80%          |
> | MATH500 | 81%      | 96.6%       |
>
> Using Qwen 2.5 1.5B Instruct as prior
>
> | Dataset | Accuracy | $f(z)$      |
> |---------|----------|-------------|
> | GSM8K   | 78%      | 82%         |
> | MATH500 | 81.4%    | 95%         |
>
> These results confirm that the prior model only needs to be able to *rewrite* the student’s reasoning into the desired style,
>
>
>
> #### **Why This Works**
>
> This behavior is consistent with prior literature [1], which shows that the quality of the training dataset strongly determine model behavior
>
> From the fact that Even weaker models can generate useful supervision signals if the task is *rewriting* rather than *problem solving*. Our method provides a **principled way to construct such high-quality SFT datasets**, enabling the student model to learn monitorable reasoning without relying on a more capable teacher.
>
> *References*
>
> 1. Muennighoff, Niklas, et al. "s1: Simple test-time scaling." Proceedings of the 2025 Conference on Empirical Methods in Natural Language Processing. 2025.
>
>
>
> > **Weakness 2:**  The faithfulness evaluation depends on judgments made by the llm-as-a-judge approach, lacking objective annotation or inter-rater reliability verification.
> The authors define monitorability as consisting of faithfulness and conciseness, but do not provide a unified quantitative formulation for this concept.
>
> **Response to Weakness 2:**  Refer to the Response to Weakness 2 of Reviewer tRVN
>
>
> > **Weakness 3:**  At line 171, the authors used 950 as a threshold to judge the conciseness. I wonder why 950?
>
> **Response to Weakness 3:** We chose the threshold **950 tokens** because it is the **shortest CoT length** at which the model still retains **>96%** of the base accuracy on MATH500. Below this length, accuracy drops sharply.
> This threshold is dataset-specific. e.g., for GSM8K, the minimum stable length is **~125 tokens**.

---

> ### Author Response · Authors · 2025-11-29
>
> > **Weakness 4:**  The experimental design could be improved by introducing more datasets. Testing on MATH500/GSK8K and MMLUPro is not sufficient. Existing models are powerful on more complex tasks like AIME, LiveCodeBench. It is better to test the method on more complex tasks.
>
> **Response to Weakness 4:** We agree with the reviewer that testing on more challenging datasets is valuable. Accordingly, we extended our evaluation to **AIME 2024**, a benchmark requiring substantially longer and more complex multi-step reasoning.
>
> We tested the trained model on AIME2024 with $\beta = 4700$
> #### Base Model
>
> | Dataset  | Base Accuracy | \( f(z) \) |
> |----------|---------------|------------|
> | AIME2024 | 30.0%         | 10%        |
>
> #### Our Fine-Tuned Model (concise, monitorable CoT)
>
> | Dataset  | Accuracy | \( f(z) \) |
> |----------|----------|-------------|
> | AIME2024 | 26.0%    | 100%        |
>
> These results show that our method can handle **complex, long-horizon reasoning**:
> we achieve **near-baseline accuracy** while producing  monitorable chains-of-thought for AIME-level tasks.

---

### Official Review · Reviewer_ArwN · 2025-11-01

**Soundness:** 2
**Presentation:** 2
**Contribution:** 2
**Rating:** 2
**Confidence:** 5

**Summary:**

This paper proposes a framework for improving the monitorability of chain-of-thought (CoT) reasoning in large language models.
The authors define two key criteria—faithfulness (alignment between reasoning and actual model decision process) and conciseness (brevity of explanations)—and introduce a prior-guided distillation method.
In this approach, a stronger teacher model (e.g., Qwen2.5-Instruct) rewrites verbose CoT traces produced by a smaller base model into shorter, more faithful forms. The refined data are then used for supervised fine-tuning of the base model.
Experiments on GSM8K, MATH500, and MMLU-Pro show that the distilled model generates shorter reasoning traces while maintaining comparable answer accuracy, suggesting improved monitorability.

**Strengths:**

1. Clear problem motivation (improving interpretability and faithfulness of CoT).

2. Well-written and reproducible experiments.

3. Results show meaningful improvement in conciseness and surface-level faithfulness without major accuracy loss.

4. Framing the goal of “monitorable reasoning” provides a useful vocabulary for reasoning safety discussions.

**Weaknesses:**

1. The method is essentially a simplification distillation—a teacher rewrites long CoTs and the student learns to mimic them.
This process is widely adopted and not novel as a training paradigm.

2. Simplifying CoTs may harm reasoning fidelity in complex multi-step problems; longer chains often encode necessary intermediate logic.
The paper does not test this on genuinely complex reasoning tasks (e.g., proofs, multi-hop logical reasoning, or coding).

3. The “faithfulness” metric relies on external LLM judgments and does not truly measure causal reasoning alignment.

**Questions:**

1. How does the proposed distillation perform on genuinely long or compositional reasoning tasks, where simplification may remove necessary steps?

2. Does the method generalize without a stronger teacher model for rewriting?

3. How is this approach different in substance from previous CoT distillation or self-distillation works (e.g., Self-Review)?

---

> ### Author Response · Authors · 2025-11-29
>
> > ** Weakness 1:**  The method is essentially a simplification distillation—a teacher rewrites long CoTs and the student learns to mimic them. This process is widely adopted and not novel as a training paradigm.
>
> **Response to Weakness 1:** We thank the reviewer for the comment. While our approach involves a teacher–student setup, it **differs fundamentally from classical distillation** in several important ways:
>
> - **The teacher is not a stronger model and cannot answer the questions.**
>    Unlike typical distillation pipelines where a more capable model provides full solutionsm, the teacher in our method is *not* solving the task. In fact, as demonstrated in our results, the teacher performs **worse** than the student on reasoning benchmarks. Thus, the teacher is *not* providing correctness or solution quality.
>
> - **The teacher does not generate full responses.**
>    The teacher only produces a *rewritten chain-of-thought*, not the final answer. The student must still:
>    - complete the reasoning, and
>    - generate the final answer independently.
>    This is not imitation of the teacher’s reasoning process, but **structured transformation learning**, where the teacher provides  constraint adhered CoT  rather than task solutions.
>
> - **Our novelty lies in the *principled dataset construction* for monitorability.**
>    Existing distillation work focuses on solution accuracy or performance improvement.
>    In contrast, our contribution is a **method for generating high-quality, monitorable CoT data** by:
>    - transforming unmonitorable reasoning traces into monitorable ones,
>    - applying a **likelihood-based rejection sampling criterion** to filter for high-quality rewrites,
>    - ensuring the rewritten traces maintain semantic alignment with the original reasoning.
>
>    This produces datasets  optimized for **monitorability**, which is not addressed by standard distillation frameworks.
>
>
>
> > **Weakness 2:** Simplifying CoTs may harm reasoning fidelity in complex multi-step problems; longer chains often encode necessary intermediate logic. The paper does not test this on genuinely complex reasoning tasks (e.g., proofs, multi-hop logical reasoning, or coding).
>
> **Response to Weakness 2:** We appreciate the reviewer’s concern  whether conciseness harms reasoning fidelity on *genuinely complex* tasks. To address this,  we conducted additional experiments on **AIME 2024**, which requires multi-step numerical reasoning with long intermediate chains.
>
> AIME is especially relevant because the **minimum effective CoT length** for solving these problems is known to be relatively large (≈4700 tokens). We therefore tested whether our conciseness-oriented training would reduce reasoning fidelity when compressing CoTs toward this lower bound.
>
> At the  Base Model evaluation at $\beta = 4700$
>
> | Dataset  | Base Accuracy | $f(z)$     |
> |----------|---------------|------------|
> | AIME2024 | 30.0%         | 10%        |
>
> After SFT training with  our proposed  dataset curation method:
>
> | Dataset  | Accuracy | $f(z)$     |
> |----------|----------|-------------|
> | AIME2024 | 26.0%    | 100%        |
>
>
>
> Our results demonstrate that, We can compress CoTs down to the minimal reasoning length needed for AIME-level problems without significantly degrading the model’s problem-solving accuracy**, while dramatically improving monitorability from 10% → 100%**.
>
> These results directly counter the concern that CoT compression degrades reasoning fidelity in complex settings. Our method reliably extracts the shortest monitorable chain-of-thought sufficient for solving the task, ensuring that the transformation from $z$ to $z_s$ preserves all essential  steps while eliminating redundancy and verbosity.
>
>
> > **Weakness 3 :** The “faithfulness” metric relies on external LLM judgments and does not truly measure causal reasoning alignment.
>
> **Response to Weakness 3:** We thank the reviewer for raising this concern. As defined in Turpin et al. (2023) [1], the use of *systematic unfaithfulness* (e.g., hint injection) is a **standard and well-established evaluation protocol** for assessing faithfulness in chain-of-thought models. This methodology has been widely adopted in recent safety and reasoning literature. Notably, Anthropic’s evaluation in [4] follows a  similar setup.
>
> *References*
> 1. Turpin, Miles, et al. "Language models don't always say what they think: Unfaithful explanations in chain-of-thought prompting." Advances in Neural Information Processing Systems 36 (2023): 74952-74965.
> 2. Baker, Bowen, et al. "Monitoring reasoning models for misbehavior and the risks of promoting obfuscation." arXiv preprint arXiv:2503.11926 (2025).
> 3. Lynch, Aengus, et al. "Agentic Misalignment: How LLMs Could Be Insider Threats." arXiv preprint arXiv:2510.05179 (2025).
> 4.  Chen, Yanda, et al. "Reasoning Models Don't Always Say What They Think." arXiv preprint arXiv:2505.05410 (2025).

---

> ### Author Response · Authors · 2025-11-29
>
> > **Question 1:** How does the proposed distillation perform on genuinely long or compositional reasoning tasks, where simplification may remove necessary steps?
>
> **Response to Question 1**   Please refer to the solution in **Response to Weakness 2**
>
> > **Question 2:** Does the method generalize without a stronger teacher model for rewriting?
>
> **Response to Question 2**   We thank the reviewer for  pointing this out. Unlike other distillation methods, we dont require a stronger teacher model. We reuse the base models capabilities to generate its CoT and transforming it into desired CoT using any prior. Our approach novelty lies in the way of converting the CoT
>
>
> **Evidence: A Much Smaller Teacher Works Equally Well**
>
> To further demonstrate that a strong teacher is unnecessary, we repeated the conciseness experiment using an even smaller model: **Qwen 2.5 1.5B Instruct**.
>
> Its standalone performance is significantly weaker:
>
> | Model                    | Accuracy | Avg. Tokens |
> |--------------------------|----------|-------------|
> | Qwen 2.5 1.5B Instruct   | 52.4     | 984         |
>
> Despite this, when used as the teacher for rewriting, the final student achieves **nearly identical results** to the 7B-teacher version.
>
>
> #### **Conciseness Experiment Results**
>
> Using Qwen 2.5 7B Instruct as Teacher (from the main paper)
>
> | Dataset | Accuracy | $f(z)$ |
> |---------|----------|-------------|
> | GSM8K   | 79%      | 80%          |
> | MATH500 | 81%      | 96.6%       |
>
> Using Qwen 2.5 1.5B Instruct as Teacher
>
> | Dataset | Accuracy | $f(z)$      |
> |---------|----------|-------------|
> | GSM8K   | 78%      | 82%         |
> | MATH500 | 81.4%    | 95%         |
>
> These results confirm that:
>
> - The teacher model **does not need to be strong**,
> - It only needs to be able to *rewrite* the student’s reasoning into the desired style,
> - A much smaller teacher yields **comparable** performance when dataset size is held constant.
>
> Our method does *not* depend on the teacher’s ability to solve problems or produce faithful reasoning.
> Instea the teacher receives the student’s original reasoning trace $z$,  and **transforms it into a more concise/monitorable version  $z_s$**.
>
> Thus, the teacher acts as a *rewriter*, not a *reasoner*.
> This distinction is central: **the teacher is not providing new reasoning, only modifying the student’s reasoning.**

---

### Official Review · Reviewer_tRVN · 2025-11-03

**Soundness:** 1
**Presentation:** 2
**Contribution:** 1
**Rating:** 2
**Confidence:** 4

**Summary:**

This paper addresses chain-of-thought (CoT) monitorability by focusing on faithfulness (whether reasoning reflects the true factors) and conciseness (brevity for monitoring). The authors claim standard reinforcement learning fails due to sparse rewards. They propose a prior-guided framework where a 7B auxiliary model transforms CoT traces from a 1.5B base model into monitorable versions, which are filtered and used for supervised fine-tuning. Experiments on MMLU-Pro, GSM8K, and MATH500 show approximately 10% faithfulness improvement and up to 60% length reduction while maintaining roughly 96% of base accuracy.

**Strengths:**

- Important problem for AI safety: Given the acceleration in AI capabilities, the field needs monitoring capabilities.
- Clear mathematical formulation: Constrained optimization framework (Eq. 1) provides principled setup.
- Proof-of-concept validation: Figure 3 demonstrates monitorable traces preserve task performance.

**Weaknesses:**

- Unsupported "RL fails" claim: No algorithm details, or hyperparameter specification provided. The contradiction with successful GRPO/PPO applications in practice makes me doubt the credibility of the results (DeepSeek-R1, o1).
- Faithfulness-conciseness tradeoff ignored: These objectives directly conflict but are treated independently with no joint optimization or Pareto analysis.
- Circular teacher dependency: Requires 7B model to fix 1.5B model. If prior generates faithful reasoning, why not use it directly?
- Uninspiring empirical results: 25% faithfulness means 75% unfaithfulness remains.
- Flawed evaluation: Hint injection is artificial proxy for real unfaithfulness.
- Missing critical baselines: No properly-tuned GRPO/PPO, or even strong prompting baselines.

**Questions:**

- What specific RL algorithm and hyperparameters did you use? Why only 500 steps? Properly-tuned GRPO works quite well with length penalty.
- How do you handle the faithfulness-conciseness tradeoff when optimizing jointly?
- Can you provide human evaluation for faithfulness claims?

---

> ### Author Response · Authors · 2025-11-29
>
> > **Weakness 1:** Unsupported "RL fails" claim: No algorithm details, or hyperparameter specification provided. The contradiction with successful GRPO/PPO applications in practice makes me doubt the credibility of the results (DeepSeek-R1, o1).
>
> **Response to Weakness 1:** We thank the reviewer for the detailed feedback. Below, we clarify the RL setup, hyperparameters, and the underlying reasons why improvements from GRPO may appear limited in our setting.
>
> #### Implementation Details
> We used the official GRPO implementation from OpenRLHF [1], a widely adopted and reliable RLHF framework. All hyperparameters used in our experiments are listed below for completeness:
>
> | Argument                         | Value                                            |
> |----------------------------------|--------------------------------------------------
> | micro_train_batch_size           | 4                                                |
> | per_device_train_batch_size      | 16                                               |
> | max_samples                      | 3200                                             |
> | max_epochs                       | 1                                                |
> | prompt_max_len                   | 1024                                             |
> | generate_max_len                 | 16384                                            |
> | zero_stage                       | 3                                                |
> | bf16                             | True                                             |
> | actor_learning_rate              | 5e-5                                             |
> | init_kl_coef                     | 0.005                                            |
> | prompt_data                      | datasets/compressed_dataset.                     |
> | apply_chat_template              | True                                             |
> | normalize_reward                 | True                                             |
> | packing_samples                  | True                                             |
> | gradient_checkpointing           | True                                             |
> | num actor GPUs                   | 4                                                |
> | gradient accumulation steps      | 4                                                |
>
>
> Dataset. We used the training split from Arora & Zanette [2], following the same curation protocol as in our main experiments.
> Evaluation. Models were evaluated every 100 RL update steps.
>
>
> #### Why We Do Not Expect Large RL Gains in Our Setting
>
> The reviewer notes that GRPO/PPO-based algorithms (e.g., DeepSeek-R1, OpenAI o1) show strong improvements in practice and raises concerns about our statement that “RL fails.” However, this interpretation overlooks a key distinction emphasized directly in the DeepSeek-R1 paper. In Section [3] 2.2.4, the authors clearly articulate the limitations of the **R1-Zero** variant:
>
> > *Although DeepSeek-R1-Zero exhibits strong reasoning capabilities and autonomously develops unexpected and powerful reasoning behaviors, it faces several issues. For instance, DeepSeek-R1-Zero struggles with challenges like poor readability and language mixing. To make reasoning processes more readable and share them with the open community, we explore DeepSeek-R1, a method that utilizes RL with human-friendly cold-start data.*
>
> This passage highlights two crucial facts:
>
> - **R1-Zero alone produces low-quality, unreadable reasoning traces.**
> - **DeepSeek-R1 succeeds only after a substantial cold-start SFT phase** built on high-quality reasoning demonstrations.
>
> From [3] Section 2.3.1 (*Cold Start*), it is clear that the **first and essential phase** of the successful DeepSeek-R1 model is **Supervised Fine-Tuning (SFT)** on a large, curated, high-quality dataset. Our work proposes a  principled approach to construct an SFT dataset specifically designed to improve monitorability.
>
> *References*
> 1. Hu, Jian, et al. "Openrlhf: An easy-to-use, scalable and high-performance rlhf framework." arXiv preprint arXiv:2405.11143 (2024).
> 2.  Arora, Daman, and Andrea Zanette. "Training language models to reason efficiently." arXiv preprint arXiv:2502.04463 (2025).
> 3. Guo, Daya, et al. "Deepseek-r1: Incentivizing reasoning capability in llms via reinforcement learning." arXiv preprint arXiv:2501.12948 (2025).

---

> ### Author Response · Authors · 2025-11-29
>
> > **Weakness 2:** Faithfulness-conciseness tradeoff ignored: These objectives directly conflict but are treated independently with no joint optimization or Pareto analysis.
>
> **Response to Weakness 2:** Thank you for this thoughtful questions. We note that faithfulness and conciseness are not inherently conflicting objectives in our setting. These properties are mutually exclusive.
>
>  - Faithfulness requires that reasoning steps logically support the conclusion. It does not mandate verbose explanations or redundant information.
>  - Conciseness requires removing unnecessary verbosity. It does not demand removing relevant reasoning steps.
>
> True to our knowledge, there is no evidence in the literature, if two things are entangled but we will check this direction in more details and improve the discussion in our work.

---

> ### Author Response · Authors · 2025-11-29
>
> **Response to Weakness 3** : This is a good point. Below, we clarify why this is **not** the case and provide empirical results demonstrating that our method does *not* rely on a strong teacher for correctness.
>
>
> #### The Teacher Does *Not* Outperform the Student in Reasoning Quality
>
> The teacher model we use, **Qwen 2.5 7B Instruct**, does *not* generate higher-reward or higher-accuracy solutions than the target student model **DeepSeek-R1 Qwen 1.5B**. To illustrate this, we evaluate the 7B teacher on **MATH500**:
>
> | Model                  | Accuracy | Avg. Tokens |
> |------------------------|----------|-------------|
> | Qwen 2.5 7B Instruct   | 76.4     | 687         |
>
> This demonstrates that the teacher is *not* providing superior reasoning or correctness. Therefore, we are **not** relying on a stronger model to “fix” a weaker one.
>
>
> **We do *not* Use the Teacher to Generate Correct Answers.** Our method does *not* depend on the teacher’s ability to solve problems or produce faithful reasoning.   Instead:
>
> - the teacher receives the student’s original reasoning trace \( z \),
> - and **transforms it into a more concise/monitorable version \( z_s \)**.
>
> Thus, the teacher acts as a *rewriter*, not a *reasoner*.   This distinction is central: **the teacher is not providing new reasoning, only modifying the student’s reasoning.**
>
>
>
> > **Weakness 4:** Uninspiring empirical results: 25% faithfulness means 75% unfaithfulness remains.
>
> **Response to Weakness 4:** We appreciate the reviewer’s concern. However, it is essential to contextualize faithfulness within the current state of the field. **Faithfulness remains a widely recognized, unsolved challenge**, even for today’s most capable frontier models. Recent work from Anthropic [5] reports that:
>
> - **Reasoning models** such as Claude 3.7 Sonnet and DeepSeek R1, and
> - **Non-reasoning models** such as Claude 3.5 Sonnet and DeepSeek V3,
>
> all exhibit **very low faithfulness rates**, often far below intuitive expectations. Thus, low faithfulness is a *systemic limitation of current LLMs*, not specific to our method or model scale.
>
> Moreover, [6] emphasizes that **monitorable chain-of-thought (CoT)** is an important prerequisite for AI safety, making even incremental improvements meaningful. In this context, our method improves the model’s faithfulness from **15% to 25%**, corresponding to a **66% relative gain**. While the absolute values remain modest—consistent with the difficulty of the task—this represents a **substantial improvement over the baseline** and demonstrates the practical utility of our dataset-construction framework.
>
> In summary, our contribution is in proposing a **principled methodology for improving CoT faithfulness and monitorability**, rather than claiming to fully solve the faithfulness problem. We view this work as an important as first step towards developing  high CoT  monitorable models.
>
> *References*
> 5.  Chen, Yanda, et al. "Reasoning Models Don't Always Say What They Think." arXiv preprint arXiv:2505.05410 (2025).
> 6.  Korbak, Tomek, et al. "Chain of thought monitorability: A new and fragile opportunity for ai safety." arXiv preprint arXiv:2507.11473 (2025).

---

> ### Author Response · Authors · 2025-11-29
>
> > **Weakness 5:** Flawed evaluation: Hint injection is artificial proxy for real unfaithfulness.
>
> **Response to Weakness 5:** We thank the reviewer for raising this concern. As defined in Turpin et al. (2023) [7], the use of *systematic unfaithfulness* (e.g., hint injection) is a **standard and well-established evaluation protocol** for assessing faithfulness in chain-of-thought models. This methodology has been widely adopted in recent safety and reasoning literature. Notably, Anthropic’s evaluation in [5] follows a  similar setup.
>
> We agree that faithfulness evaluation is an evolving area, and we welcome improvements.
>
> *References*
> 7. Turpin, Miles, et al. "Language models don't always say what they think: Unfaithful explanations in chain-of-thought prompting." Advances in Neural Information Processing Systems 36 (2023): 74952-74965.
> 8. Baker, Bowen, et al. "Monitoring reasoning models for misbehavior and the risks of promoting obfuscation." arXiv preprint arXiv:2503.11926 (2025).
> 9. Lynch, Aengus, et al. "Agentic Misalignment: How LLMs Could Be Insider Threats." arXiv preprint arXiv:2510.05179 (2025).
>
> > **Weakness 6:** Missing critical baselines: No properly-tuned GRPO/PPO, or even strong prompting baselines.
>
> **Response to Weakness 6:**  We thank the reviewer for pointing this out. We clarify below that:
>   - As detailed in our **Response to Weakness 1**, we use the GRPO implementation and hyperparameters from OpenRLHF, a widely adopted RLHF framework.
>   - We did include strong prompting baselines for  faithfulness in our paper, Fig 4

---

> ### Author Response · Authors · 2025-11-29
>
> > **Question 1:**  What specific RL algorithm and hyperparameters did you use? Why only 500 steps? Properly-tuned GRPO works quite well with length penalty.
>
> **Response to Question 1**  We present the RL algorithm and Hyperparameter in detail in the **Response to Weakeness 1.**
>
> > **Question 2:**  How do you handle the faithfulness-conciseness tradeoff when optimizing jointly?
>
> **Response to Question 2:**  Refer to  Response to Weakness 2
>
> > **Question 3:** Can you provide human evaluation for faithfulness claims?
>
> **Response to Question 3:** For evaluating faithfulness, we employ an **LLM-as-a-judge** setup using Qwen2.5-14B-Instruct. We selected this model because it provides strong language understanding while remaining feasible within our compute budget. This evaluation strategy is widely used and increasingly standard in the literature: both OpenAI’s [8] and Anthropic’s [9] rely on LLM judges to assess reasoning quality and evaluate model behavior. Human evaluations are a valid scope of future research.

---

### Note · Authors · 2025-11-29

I have read and agree with the venue's withdrawal policy on behalf of myself and my co-authors.